# Defined Pig Microbiota with a Potential Protective Effect against Infection with *Salmonella* Typhimurium

**DOI:** 10.3390/microorganisms11041007

**Published:** 2023-04-12

**Authors:** Kristyna Horvathova, Nikol Modrackova, Igor Splichal, Alla Splichalova, Ahmad Amin, Eugenio Ingribelli, Jiri Killer, Ivo Doskocil, Radko Pechar, Tereza Kodesova, Eva Vlkova

**Affiliations:** 1Department of Microbiology, Nutrition and Dietetics, Faculty of Agrobiology, Food and Natural Resources, Czech University of Life Sciences Prague, 165 00 Prague, Czech Republic; horvathovak@af.czu.cz (K.H.); modrackova@af.czu.cz (N.M.); ahmadamin@af.czu.cz (A.A.); ingribelli@af.czu.cz (E.I.); killer@iapg.cas.cz (J.K.); doskocil@af.czu.cz (I.D.); pecharr@af.czu.cz (R.P.); kodesovat@af.czu.cz (T.K.); 2Laboratory of Gnotobiology, Institute of Microbiology, Czech Academy of Sciences, 549 22 Novy Hradek, Czech Republic; splichal@gnotobio.cz (I.S.); splichalova@gnotobio.cz (A.S.); 3Laboratory of Anaerobic Microbiology, Institute of Animal Physiology and Genetics, Czech Academy of Sciences, 142 20 Prague, Czech Republic

**Keywords:** pig intestinal bacteria, bacterial consortium, probiotic properties testing, intestinal pathogens, gnotobiotic piglets

## Abstract

A balanced microbiota is a main prerequisite for the host’s health. The aim of the present work was to develop defined pig microbiota (DPM) with the potential ability to protect piglets against infection with *Salmonella* Typhimurium, which causes enterocolitis. A total of 284 bacterial strains were isolated from the colon and fecal samples of wild and domestic pigs or piglets using selective and nonselective cultivation media. Isolates belonging to 47 species from 11 different genera were identified by MALDI-TOF mass spectrometry (MALDI-TOF MS). The bacterial strains for the DPM were selected for anti-*Salmonella* activity, ability to aggregate, adherence to epithelial cells, and to be bile and acid tolerant. The selected combination of 9 strains was identified by sequencing of the 16S rRNA gene as *Bacillus* sp., *Bifidobacterium animalis* subsp. *lactis*, *B. porcinum*, *Clostridium sporogenes*, *Lactobacillus amylovorus*, *L. paracasei* subsp. *tolerans*, *Limosilactobacillus reuteri* subsp. *suis*, and *Limosilactobacillus reuteri* (two strains) did not show mutual inhibition, and the mixture was stable under freezing for at least 6 months. Moreover, strains were classified as safe without pathogenic phenotype and resistance to antibiotics. Future experiments with *Salmonella*-infected piglets are needed to test the protective effect of the developed DPM.

## 1. Introduction

The gastrointestinal (GI) tract represents the most extensive interface between the host, environmental factors, and antigens in the human body. During the lifetime, a large amount of food passes through the digestive tract, together with microorganisms from the environment, which pose a considerable threat to the integrity of the gut [1]. The GI microbiota represents trillions of microbes adhering to the intestinal wall or occurring loosely in the lumen of the intestine [2]. It is highly individual [3] and is composed of more than 4000 different prokaryotes [4], but also viruses, archaea, and eukarya are present [5]. Furthermore, the GI microbiota has co-evolved with the host [6] to form a complex symbiotic relationship ranging from mutualism to parasitism [7,8].

GI microbiota has many functions, including converting indigestible food residues, production of short-chain fatty acids and vitamins, stimulating immune responses, and formation of barriers against pathogenic microorganisms [9]. Commensal bacteria can affect the host’s health beneficially or detrimentally [10]. For example, a balanced microbiota promotes resistance to intestinal pathogens [11] and protects against overgrowth with indigenous pathobionts (opportunistic pathogens) [12,13]. On the other hand, pathogens developed strategies to escape colonization resistance mediated by commensal microorganisms and used them for their profit [14,15]. Therefore, the interplay between pathogens, commensals, or indigenous pathobionts is decisive for controlling infectious diseases. Understanding the interactions between pathogens and commensal microorganisms can lead to new therapeutic approaches and the treatment of infectious diseases [12].

Among the most common intestinal pathogens worldwide belong to *Salmonella enterica* serovar Typhimurium (*S. typhimurium*), which affects both the small and large intestines [16] and causes enterocolitis in humans, pigs, and other warm-blooded animals [17]. Annually 200,000 people die from salmonellosis caused by non-typhoidal *Salmonella* spp. It is typically characterized by a self-limiting gastroenteritis syndrome expressed by anorexia, diarrhea, fever, and vomiting [18]. *S. typhimurium* also can cause life-threatening illnesses in immunocompromised hosts [19]. In swine, *Salmonella* strains cause septicemia, enterocolitis, or subclinical infections [20]. The enterocolitis form associated with *S. typhimurium* has low mortality and high morbidity [21], with the signs described above. Subclinical forms do not cause disease but may be associated with reduced productivity and average daily gain [22], and increased risk of food contamination [23]. Livestock farms, in general, are a significant source of salmonellosis [24]. Feed antibiotics, banned in the European Union [25], were used to limit intestinal infections in farm animals as prevention in the past [26]. One of the promising possibilities to suppress infections with enteric pathogens, and favorably influence the composition of the intestinal microbiota and stimulate the host immune response, can be the use of defined multispecies synthetic microbiota composed of host-specific strains. Such microbiota support colonization resistance as a result of mutual interactions between the host and its original microbes and microbes and each other [27,28,29]. Probiotics are considered to be well-tolerated and safe and are selected to exhibit potential health benefits to the host through modulation of the gut microbiota [30], e.g., as the prevention of nosocomial [31] and antibiotic-associated diarrhoea [32], management of acute gastroenteritis symptoms [33], reduction of the eczema incidence [34], alleviation of lactose intolerance [35], and protection against gut dysbiosis associated with learning and memory disorders [36]. Beneficial effects are exerted through a variety of mechanisms, including the production of bacteriocins, lowering gut pH and colonization, invading pathogenic organisms, and modifying the immune response, viability, and activity of passage through the GI tract [37] where probiotics must survive the unfavorable conditions of the environment [30]. Commercial probiotics with a non-specific effect on the host are often used in animal and human nutrition [38].

To select new probiotic strains with specific and required properties, in vitro assays are commonly used as a screening tool to identify potential probiotic properties [39,40,41]. Probiotics’ functionality is then verified using in vivo systems with laboratory or farm animals [42]. Interestingly, there are the most often used mice experiments [42] because the availability of mutant and genetically modified mouse strains greatly facilitates functional studies [43]. These studies use human bacteria to colonize gnotobiotic mice, although there are phylogenetic differences between them. The human [44] and mouse [45] microbiota are divergent, and they can influence microbe–host interactions and the long-term stability of colonization [46]. In contrast, human [44] and pig [47] microbiomes show high similarities. Moreover, pigs show close anatomic, physiologic, and genetic similarities to humans, and results obtained from pig models should be more plausibly translated to humans [48].

Our study aimed to screen the anti-*Salmonella* potential of pig bacterial isolates and develop probiotic-defined pig microbiota with functional and technologically desired properties for future use as a novel protective strategy against the infection with *Salmonella* Typhimurium in in vivo systems.

## 2. Materials and Methods

### 2.1. Bacterial Strain Isolation

The broad spectra of commensal bacteria were isolated from colon and fecal samples of pigs/piglets bred in Czech farms (*n* = 25, samples taken at slaughterhouses) and wild boars (*n* = 25, samples collected during hunting). Samples were collected in tubes containing 5 g/L tryptone, 5 g/L nutrient broth No. 2, 2.5 g/L yeast extract (all Oxoid, Basingstoke, UK), 0.5 g/L L-cysteine, 1 mL/L tween 80 (both Sigma-Aldrich, St. Louis, MO, USA), and 30% glycerol (V.W.R., Radnor, PA, USA). After the sample collection and transfer to the place of analysis, log 10 serial dilutions of samples were plated on different media cultivated under various cultivation conditions (Table 1).

Bacterial isolates were obtained based on different cultivation characteristics of colonies from all used media. The oxygen, temperature, and substrate requirements were secured for the following sub-cultivation. Anaerobes were cultured in Wilkins–Chalgren Anaerobe broth supplemented with soya pepton (5 g/L; both Oxoid), L-cysteine (0.5 g/L), and tween 80 (1 mL/L, both Sigma-Aldrich), while aerobes in Tryptone Soya broth (Oxoid), both for 24 h. Then, bacterial isolates were stored as stock cultures at −80 °C in 30% glycerol.

### 2.2. Culture Identifications

Bacterial isolates were identified by MALDI-TOF mass spectrometry (MALDI-TOF MS) with ethanol-formic acid extraction procedure with HCCA matrix solution according to the manufacturer’s instructions (Bruker Daltonik GmbH, Bremen, Germany) with the usage of an extended custom database in Biotyper software (Bruker) for identification of bifidobacteria [50].

Based on species variability, the selected isolates were then identified by 16S rRNA gene amplicon sequencing. DNA was isolated using PrepMan Ultra™ (Applied Biosystems, Waltham, MA, USA) according to the manufacturer’s instructions. For PCR amplifications of the 16S rRNA gene, the universal pair of primers fD1/rP2 [51] was used, except for pair 285F/261R [52] for the identification of bifidobacteria. PCR products were purified by the EZNA Cycle Pure Kit (Omega Bio-Tek, Norcross, GA, USA) and sequenced by Eurofins Genomics (Ebersberg, Germany). The obtained sequences were processed in Chromas Lite 2.5.1 (Technelysium Pty Ltd., Tewantin, Australia), BioEdit [53] with ClustalW algorithm [54], and compared with 16S rRNA gene sequences in BLAST rRNA/ITS (https://blast.ncbi.nlm.nih.gov/https://www.ezbiocloud.net/, accessed on 4 November 2022) and EZBioCloud databases (https://www.ezbiocloud.net/, accessed on 4 November 2022).

### 2.3. Characterization of In Vitro Potentially Probiotic Properties

The antimicrobial activity of cell-free supernatants from all isolated pig commensal bacteria was tested against streptomycin-sensitive wild-type *Salmonella* Typhimurium LT2 [55] and its streptomycin-resistant mutant (STM) [56] on Tryptone Soya agar (Oxoid) by the agar-well diffusion method in triplicates. In the second step, the activity of neutralized supernatant was determined [57].

Isolates were grouped into two phenotypes in relation to auto-aggregation ability (Agg). Strains were forming sand-like particles by aggregated cells gravitating to the bottom of tubes (if cultivated in liquid media), resulting in a clear solution of media (Agg+) and bacteria without Agg ability (Agg−), producing constant turbidity for a long time. Moreover, Agg ability was expressed as a percentage (%) after the procedure described elsewhere [58].

Properties predicting the survival of commensal bacteria during passage through the animal GI tract were tested using the method described by [59]. After the incubation of tested strains in PBS buffer adjusted with HCl (Sigma-Aldrich) to pH 2 or containing 1.5% of bile extract (Oxoid), the residual viable counts were determined by standard plate count on appropriate media. Then, the antibiotic resistance/sensitivity assay was performed using a disc diffusion method under conditions determined by EUCAST [60]. For the interpretation of zone diameters, the breakpoint tables were used (Version 11.0, 2021, http://www.eucast.org). Phenotypic tests associated with a pathogenic potential of *Bacillus* spp. were performed. Hemolytic activity was determined by plating a freshly grown culture on Columbia Blood agar (Oxoid) supplemented with 5% sheep blood (*v*/*v*, Oxoid) according to [61]. To assess lecithinase activity, the freshly grown cultures were plated on Tryptone Soya agar enriched with a sterile egg yolk emulsion (100 mL/L, Oxoid) [62].

### 2.4. Adherence to the Porcine IPEC-J2 Cell Line

Strains with Agg phenotype and representatives from isolated genera were tested for the adhesion to intestinal cell lines in triplicates. The bacterial adhesion methods previously described by [63] were used with some modifications. The intestinal porcine epithelial IPEC-J2 (ACC 701) cell lines were obtained from the German Collection of Microorganisms and Cell Cultures (DSMZ, Braunschweig, Germany). For the visualization of adhered bacteria, the fluorescein isothiocyanate (Life Technologies, Carlsbad, CA, USA) was used with a final concentration 25 μg/mL. The staining procedure took 30 min at 37 °C in the dark. Fluorescence was then measured using Tecan Infinite M200 (Tecan Group, Männedorf, Switzerland) at an excitation wavelength of 495 nm and emission wavelength of 519 nm. The percentage (%) of adherent bacteria was calculated as described elsewhere [64].

### 2.5. Preparation of Defined Pig Microbiota (DPM) and Its Stability

The mutual inhibition of the strains selected for DPM was tested by the agar-well diffusion method as described above. Then, the strains were co-cultured in appropriate broth, and their numbers were determined by the standard plate counts technique after the cultivation in the mixture or as a single strain. Overnight bacterial cultures were collected by centrifugation (12,000× *g*, 4 °C, 5 min), resuspended in PBS buffer, and centrifuged again to completely remove cultivation media. After that, all bacterial strains were resuspended in PBS buffer with 30% glycerol and mixed to a final concentration of 5 × 10^8^ CFU/mL (approximately 5.50 × 10^7^ CFU/mL for each strain). All procedures were carried out under anaerobic conditions (Bugbox, BioTrace, Bridgend, UK). The mixture was aliquoted into cryovials in final volume 1.2 mL (6 × 10^8^ CFU/mL), and single doses of stock cultures were stored at −28 °C and −80 °C. Bacterial mixture stability was determined by cultivation on media described in Table 1, after 1 day, then 1, 3, and 6 months of storage. Three single doses were used for each determination. The identity of re-isolated strains was verified by MALDI-TOF MS [50].

### 2.6. Statistics

Bacterial mixture stability and viability (cultivation counts in log CFU/mL) were evaluated using StatSoft, Inc. (2013) (STATISTICA—data analysis software system, version 12, www.statsoft.com) and MS Excel (Redmond, WA, USA). The normality of the data was assessed by the Shapiro–Wilk W test (α = 0.05). Differences within the storage temperature were evaluated using a *t*-test (α = 0.05), and differences within the time duration of storage by one-way ANOVA (α = 0.05).

### 2.7. Ethical Approval

The work with animals was conducted according to the ethical standards defined by the EU legislation on the use of experimental animals (2010/63/EU) and approved by the Animal Care and Use Committee of the Czech Academy of Sciences (protocol 57/2021; 18 August 2021).

## 3. Results

### 3.1. Bacterial Isolation and Identification

A total of 284 bacterial strains were isolated from the colon and fecal samples of wild and domestic pigs or piglets. The isolates belonged to 47 species from 11 different genera as determined by MALDI-TOF MS (Table 2).

The identity of strains selected for DPM was verified by the 16S rRNA gene sequencing (Table 3). Bacteria were identified as *Bacillus* sp. (*Bacillus paramycoides/paranthracis*/*nitratireducens* identification group), *Bifidobacterium animalis* subsp. *lactis*, *Bifidobacterium porcinum*, *Clostridium sporogenes*, *Lactobacillus amylovorus*, *Lactobacillus paracasei* subsp. *tolerans*, *Limosilactobacillus reuteri* subsp. *suis*, and the next two *Limosilactobacillus reuteri* strains that were further indistinguishable at the subspecies level (*kinnaridis/murium/porcinus/rodentium/suis*) by used methods.

### 3.2. Characterization of In Vitro Potentially Probiotic Properties

One hundred and fifteen cell-free supernatants inhibited the growth of *Salmonella* strains. Inhibition zones ranged from 7.00 to 13.33 mm, 9.23 ± 1.29 mm on average (Table 3 and Appendix A). The most active isolates belonged to *Lactobacillaceae* family and genera *Bifidobacterium*, *Enterococcus*, and *Clostridium*. No anti-*Salmonella* activity was observed for strains of genera *Acidaminococcus*, *Bacillus*, *Bacteroides*, *Escherichia*, *Paeniclostridium*, *Paraclostridium*, *Streptococcus*, *Staphylococcus*, and *Terrisporobacter*. The pH values of active cell-free supernatants varied between 3.5 and 4.1. The neutralized supernatants did not inhibit tested *Salmonella* strains. The pH of non-active supernatants reached neutral or slightly acid values.

Eighty strains with anti-*Salmonella* activity and/or the ability to adhere to epithelial cells were tested for the properties predicting their survival ability in the gastrointestinal tract (Table 3 and Appendix A). In general, the tested bacteria were more sensitive to 1.5% bile extract than to low pH, but obtained results are highly strain-specific. Forty-two of 80 strains tested showed excellent bile tolerance (<1 log CFU/mL decrease in viability) after 3 h of incubation. Twenty-two isolates showed a decrease in viability even lower than 0.25 log CFU/mL. The survival of commensal bacteria at low pH was slightly better than in 1.5% bile extract. Fifty-one showed a decrease in viability lower than 1 log CFU/mL at pH 3 after 2 h of incubation and counts of 26 strains did not decrease under 0.25 log CFU/mL.

None of the tested strains was resistant to antibiotics, according to EUCAST breakpoint tables. After the cultivation of *Bacillus* spp. on agar containing blood, neither clear nor green zones were detected, and strains were considered non-hemolytic. Strain *B. licheniformis* PC2 showed a slight lecithinase activity after the cultivation on the agar with egg yolk. No precipitation zones were observed in the other two tested *Bacillus* strains.

### 3.3. Auto-Aggregation and Adherence to the Porcine IPEC-J2 Cell Line

Agg phenotype was just observed in some strains of *Bifidobacterium* spp. and *Lactobacillaceae* and varied between 21.83 and 78.00%. The adhesion to epithelial cell lines was predominantly determined in strains with Agg phenotype but the representatives from each isolated genus were also tested. Bacteria adhered to porcine epithelial cell line IPEC-J2 in a range from 0.21 ± 0.08% to 17.42 ± 2.22 (Table 3 and Appendix A). Agg phenotype was not always in correlation with adherence capacity.

### 3.4. Defined Pig Microbiota and Its Stability

Based on tested properties, in the final, 9 strains (Table 3) were selected for DPM preparation with a potential protective effect against *Salmonella* infections suitable for future application to gnotobiotic piglets. The bacteria did not inhibit each other in their growth as determined by the agar-well diffusion method and standard plate counts technique after their co-cultivation. The stability of the bacterial mixture was verified on 1 day and after 1, 3, and 6 months of freezing at −28 °C and −80 °C by the cultivation. An initial concentration of 5 × 10^8^ CFU/mL was maintained throughout the whole storage period with total numbers oscillating between 5.89 × 10^8^ to 7.79 × 10^8^ CFU/mL, not depending on storage temperature without statistically significant differences in both monitored parameters. The MALDI-TOF MS analysis confirmed the presence of all included bacterial species in each analyzed dose.

## 4. Discussion

There are various ways to realize microbiota-based therapy, such as fecal microbiota transplantation, diet and probiotics, synthetic microbiota, and microbiota-derived bioactive compounds [65]. A mono-association of germ-free piglets [66] with commensal pig bifidobacteria [67,68] and lactobacilli [69] mildly alleviated signs of *S. typhimurium*-caused enterocolitis. A comparison of the protective effect of three well-known probiotics *Bifidobacterium animalis* subsp. *lactis* BB-12 [70], *Lactobacillus rhamnosus* GG [71], and *Escherichia coli* Nissle 1917 [67,69] caused amelioration of the subsequent infection with *S. typhimurium*, as well. However, the protection was highly significant only in *E. coli* Nissle 1917 [69]. A more complex effect of multistrain microbiota on the host is possible to expect [72,73]. Thus, current probiotic preparations are usually composed of several bacterial species/strains.

A Bristol microbiota was derived from Altered Schaedler Flora microbiota composed of three murine bacterial species. After it was applied to hysterectomy-derived GF piglets, it was able to colonize the piglet gut without any deleterious effect on piglets [74]. However, it is known that the association of the GI tract with species-specific microbiota is more effective than microbiota derived from different species [75]. Unfortunately, commercial probiotics currently used to modify animal intestinal microbiota have non-specified effects on the host, and their colonization ability is limited. Thus, these feed supplements may not have the desired health effect [76]. Furthermore, in addition to this lack of guaranteed outcomes, there may be no less significant problem of product viability during administration in farm conditions [77]. Commonly used probiotic microbes in pig production often consists only of enterococci [78] and a few bacterial strains [79]. It would be desirable to administer a broader spectrum of microorganisms to achieve the targeted effect. Bifidobacteria [80] and lactobacilli [81], which commonly inhabit the pig intestines, thus also could be promising as probiotics for pigs. Therefore, if the aim of the probiotic application is to suppress salmonellosis, it is advisable to use a bacterium or a bacterial mixture with a proven effect against *Salmonella*. Experiments with farm animals usually consist in feeding non-specific probiotic strains and then monitoring the effect on the performance of the animals [82]. On the contrary, an approach chosen in our work was precisely targeted. We aimed to develop specific DPM from pig origin strains with the potential ability to protect piglets against infection with *Salmonella* Typhimurium.

Recently, a collection of bacterial GI commensals isolated from the pig GI tract consisting of 110 species across 40 families and nine phyla have been introduced [83]. It reveals many taxonomic groups with specific metabolic functions and probiotic potential for pigs. Similarly, we cultured a wide variety of different bacterial strains and tested their potential probiotic properties and anti-*Salmonella* effect. In our study, almost 300 isolates of 47 species belonging to 11 genera were isolated from colon and fecal samples of wild and domestic pigs or piglets. The commensals suitable for preparing a DPM with a potential protective effect against intestinal infections were chosen primarily based on their antimicrobial activity against *Salmonella* Typhimurium strains and their prerequisite for colonization of the gut. When cultivated under laboratory conditions, it was found that some strains of potentially probiotic genera *Acidaminococcus*, *Eubacterium*, and *Mitsuokella* cannot be kept alive. Therefore, further analyses with these isolates could not be performed. Our results showed that cell-free supernatant from nearly half of the tested bacteria, including *Lactobacillaceae*, *Bifidobacterium, Enterococcus*, and *Clostridium* strains, has antimicrobial activity against *Salmonella* LT2 and its streptomycin-resistant mutant. Since the neutralized cell-free supernatant did not inhibit *Salmonella* strains, antimicrobial activity is caused especially by a decrease in pH by the production of organic acids [84]. A reduction in the antagonistic effect of cell-free supernatants from potentially probiotic strains after its neutralization was also reported in the study of Tejero-Sariñena et al. [85]. However, the production of other antibacterial compounds cannot be excluded. Secretion of specific antimicrobial peptides and proteins by commensal bacteria was reported elsewhere [86,87,88]. Inhibitory activity of various putative probiotic strains against *S. typhimurium* and other intestinal pathogens was reported in many studies [89,90,91].

In synergy with the production of antimicrobial substances, colonization resistance is an important mechanism in host protection [11]. So, the bacterial consortium had to be able to adhere to the intestinal mucosa and colonize the intestines [92]. Moreover, strains with high adhesive properties may play a role in colonizing the mucosa with enteropathogens through competitive and binding sites on the intestinal epithelium, providing more health benefits. It can emphasize the local action of some metabolites that are produced by probiotic bacteria [93]. Some studies have reported that adhesion ability may correlate closely with auto-aggregation [94]. This fact was not fully confirmed in our study. There were some strains with strong auto-aggregation phenotypes which were not able to adhere to the porcine IPEC-J2 cell line. On the other hand, some isolates showing zero auto-aggregation capacity strongly adhered to epithelia. A presumption for successful colonization of the intestine is survival ability during the passage through the host’s upper parts of the GI tract [30]. In our experiment, many strains showed excellent low pH and bile tolerance and did not decrease in viability by more than 0.25 log CFU/mL. Similar results were reported by [95,96]. In general, the pig-origin strains tested in our study were slightly more sensitive to bile extract than to low pH. These results are in contrast with our previous studies, where calf- and lamb-origin strains were tested and showed higher sensitivity to low pH [57,97].

Well-identified bacterial strains for a mixture with a potentially protective effect against *Salmonella* infection were selected. Our DPM composed of 9 strains showed the following properties: either anti-*Salmonella* activity or adherence capacity, tolerance to low pH and bile extract, non-pathogenic phenotype, absence of antibiotic resistance and mutual inhibition, and stability during storage under freezing conditions for a long time duration. These characteristics are standardly used for testing bacteria with potential probiotic effects [98,99] but have only predictive value.

Probiotics are usually applied in an inactive form as spray-dried, frozen, or freeze-dried. Moreover, all these variants have been verified as suitable for preservation and distribution [100]. In the study of Geigerova et al. [101], it was proven that the preservation form of probiotics does not affect their ability to colonize the intestine. Similarly to Darnaud et al. [102], we used freezing as a preservation method for its simplicity and the long-term survival of all strains of newly developed DPM in this form.

There are numerous studies using minimally defined microbiota in gnotobiotic models [27,46]. Furthermore, well-characterized strains are usually used but without previous testing for their mutual interactions [102]. In our study, bacteria were selected according to functional properties, and their mutual interactions were emphasized. Our approach increases the probability of successful colonization of all bacteria from the consortia.

## 5. Conclusions

Based on the in vitro results, the mixture of nine commensal bacteria with anti-*Salmonella* activity was developed for possible application to piglets. Selected bacteria which were able to adhere to epithelial cells, were bile and acid tolerant, classified as safe without pathogenic phenotype and resistance to antibiotics, did not show mutual inhibition, and had favorable technological properties. Nevertheless, future experiments with gnotobiotic and conventional piglets are needed to verify the protective effect of the bacterial consortium in vivo.

## Figures and Tables

**Table 1 microorganisms-11-01007-t001:** Cultivation media and conditions used for isolation of bacteria from pig samples.

Targeted Bacterial Groups	Cultivation Medium	Cultivation Condition
Total anaerobes	Wilkins–Chalgren agar, 5 g/L soya peptone (both Oxoid), 0.5 g/L L-cysteine, 1 mL/L tween 80 (both Sigma-Aldrich), WSP agar	37 °C, 48 h, anaerobiosis
Bifidobacteria	WSP agar, 100 mg/L norfloxacin, 100 mg/L mupirocin (both Oxoid), 1 mL/L glacial acetic acid (Sigma-Aldrich) [49]	37 °C, 48 h, anaerobiosis
Sporulates	WSP agar/Tryptone Soya agar (Oxoid)	37 °C, 48 h, anaerobiosis/aerobiosis; 80 °C for 10 min prior cultivation
Lactobacilli	Rogosa agar (Oxoid), 1.32 mL/L acetic acid (Sigma-Aldrich)	37 °C, 72 h, microaerophilic conditions
Coliforms	TBX agar (Oxoid)	37 °C, 24 h, aerobiosis

**Table 2 microorganisms-11-01007-t002:** Bacterial species isolated from colon and fecal samples of pigs and piglets.

Targeted Bacterial Groups	Isolated Bacteria (Number of Isolates)	
Total anaerobes	*Acidaminococcus fermentans* (5)	*Eubacterium tenue* (3)
	*Bacteroides uniformis* (9)	*Mitsuokella mulracida* (5)
	*Enterococcus durans* (5)	*Staphylococcus aureus* (6)
	*Enterococcus faecalis* (9)	*Staphylococcus warneri* (3)
	*Enterococcus faecium* (7)	*Streptococcus alactolyticus* (2)
Bifidobacteria	*Bifidobacterium animalis* (12)	*Bifidobacterium pseudolongum* (8)
	*Bifidobacterium apri* (3)	*Bifidobacterium thermophilium* (9)
	*Bifidobacterium boum* (9)	*Mitsuokella mulracida* (6)
	*Bifidobacterium porcinum* (12)	
Sporulates	*Bacillus amyloliquefaciens* (2)	*Bacillus vallismortis* (2)
	*Bacillus cereus* (2)	*Clostridium cochlearium* (3)
	*Bacillus licheniformis* (7)	*Clostridium perfringens* (10)
	*Bacillus mycoides* (8)	*Clostridium sporogenes* (12)
	*Bacillus paramycoides* (5)	*Paeniclostridium sordellii* (8)
	*Bacillus subtilis* (8)	*Paraclostridium bifermentans* (4)
	*Bacillus thuringiensis* (1)	*Terrisporobacter glycolicus* (1)
Lactobacilli	*Lacticaseibacillus paracasei* (5)	*Lactobacillus porci* (8)
	*Lactobacillus amylovorus* (17)	*Ligilactobacillus agilis* (2)
	*Lactobacillus antri* (2)	*Ligilactobacillus ruminis* (10)
	*Lactobacillus delbrueckii* (2)	*Ligilactobacillus salivarius* (3)
	*Lactobacillus fermentum* (2)	*Limosilactobacillus mucosae* (4)
	*Lactobacillus johnsonii* (5)	*Limosilactobacillus reuteri* (18)
	*Lactobacillus kitasatonis* (2)	
Coliforms	*Escherichia fergusonii* (3)	*Shigella flexneri* (4)
	*Escherichia coli* (11)	

**Table 3 microorganisms-11-01007-t003:** Identification and functional properties of commensal bacteria isolated from pig and piglet colon and fecal sample.

				Delta log CFU/mL Decrease after Incubation in								
				pH 3		1.5% Bile		Inhibition Zones (mm) *				
Strain	Identification by 16S rRNA Gene Sequencing	GenBank	Origin	1 h	2 h	2 h	3 h	LT2	STM	Agg	Agg% **	Adhesion **
PG1	*Bacillus paramycoides/paranthracis/nitratireducens*	OP778050	domestic pig colon	0.75	0.97	0.26	0.44	6.00 ± 0.00	6.00 ± 0.00	−	nt	0.41 ± 0.19
PG2	*Bifidobacterium animalis* subsp. *lactis*	OP778043	domestic pig feces	0.17	0.19	0.00	0.10	7.67 ± 0.58	8.33 ± 0.58	+	73.66 ± 1.53	17.42 ± 2.22
PG3	*Bifidobacterium porcinum*	OP778042	domestic pig feces	0.06	0.17	0.15	0.16	9.00 ± 0.00	8.00 ± 0.00	+	41.07 ± 3.10	0.17 ± 0.08
PG4	*Clostridium sporogenes*	OP778045	domestic pig colon	0.03	0.07	0.11	0.13	7.33 ± 0.58	7.00 ± 0.00	+	24.37 ± 1.52	0.41 ± 0.10
PG5	*Lactobacillus paracasei* subsp. *tolerans*	OP778044	domestic pig colon	0.07	0.21	0.06	0.33	13.00 ± 0.00	8.33 ± 0.58	+	61.27 ± 3.61	2.29 ± 0.46
PG6	*Lactobacillus amylovorus*	OP778047	wild pig colon	0.19	0.23	0.45	0.79	10.00 ± 0.00	8.67 ± 0.58	+	21.83 ± 2.22	0.27 ± 0.03
PG7	*Limosilactobacillus reuteri* subsp. *porcinus*	OP778046	wild pig colon	0.03	0.13	0.01	0.01	9.00 ± 0.00	10.67 ± 0.58	+	54.57 ± 2.30	1.13 ± 0.64
PG8	*Limosilactobacillus reuteri* subsp. *suis*	OP778049	domestic pig feces	0.11	0.14	0.07	0.11	8.33 ± 0.58	10.00 ± 0.00	+	55.80 ± 0.44	1.63 ± 0.75
PG9	*Limosilactobacillus reuteri* subsp. *porcinus*	OP778048	domestic piglet colon	0.01	0.32	0.05	0.08	7.67 ± 0.58	7.33 ± 0.58	+	32.33 ± 1.33	0.74 ± 0.36

* Susceptibility of *Salmonella* strains to cell-free supernatant of pig origin commensals (diameters are means of three determination ± SD, diameter of the well = 6.00 mm); ** *n* = 3, mean ± SD; LT2-*S. typhimurium*; STM—*S. typhimurium*; Agg—auto-aggregation (scored positive when clearly visible sand-like particles were formed); Agg%—auto-aggregation %; nt—not tested.

## Data Availability

Data are available upon request from the corresponding author.

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
