# Peer review of "Defined Pig Microbiota with a Potential Protective Effect against Infection with *Salmonella* Typhimurium"

_microorganisms, 2023, doi:10.3390/microorganisms11041007_

Round 1
Reviewer 1 Report
Major comments
Introduction
- add more references and data about the S. Typhimurium infection in pigs(pathogenesis, losses, etc)
- L67-69: add more references and data about the use of probiotics
Materials and Methods
- provide details about the ethics (ethics committee, approval number)
Discussion :
- you could discuss the economic impact of your results on Brazilian pig production.
Conclusion:
- You should expand the part of ‘’conclusion’’
Minor comments
- L3, L18:, L91 Salmonella Typhimurium (italics)
- L24: Bifidobacterium porcinum
- L25: Lactobacillus paracasei subsp. Tolerans
- L55, 59: S. Typhimurium (italics)
- L89: probiotic-defined pig
- L191: Table 2: full name of ‘’ S. warneri’’
- L244: was verified on 1 day
Author Response
Major comments:
Introduction
- add more references and data about the S. Typhimurium infection in pigs (pathogenesis, losses, etc)
Authors comment: References and data were added (L 60-67 of the revised manuscript). Salmonella pathogenesis is also mentioned in the discussion (L 284-290 of the revised manuscript).
- L67-69: add more references and data about the use of probiotics
Authors comment: References and data were added (L 75-79 of the revised manuscript).
Materials and Methods
- provide details about the ethics (ethics committee, approval number)
Authors comment: The paragraph “ethical approval” was added to the methodology (L 201-205 of the revised manuscript).
Discussion:
- you could discuss the economic impact of your results on Brazilian pig production.
Authors comment: The discussion on economic impact of our results on pig production would be highly speculative. Application of the developed defined pig microbiota could have positive effect, but further studies are necessary to verify it. The discussion on positive effect of multi-species probiotics was extended (L 284-290 of the revised manuscript).
Conclusion:
- You should expand the part of ‘’conclusion’’
Authors comment: The conclusions part was extended (L 375-378 of the revised manuscript)
Minor comments:
- L3, L18:, L91 Salmonella Typhimurium (italics)
Authors comment: The bacterium name „Salmonella enterica serovar Typhimurium“ used in our manuscript is correct and used in accordance with the principles of nomenclature. Genus Salmonella consists of two species: Salmonella enterica and Salmonella bongori. Salmonella enterica has six subspecies and approximately 2,500 serovars. Salmonella Typhimurium is one of serovars of Salmonella enterica subsp. enterica. Thus, Salmonella is written in italics but its serovar Typhimurium is in normal font. The same role is applied to the abbreviated S. Typhimurium. Please, see Hurley et al. 2014 (Hurley, D.; McCusker, M.P.; Fanning, S.; Martins, M. Salmonella-Host Interactions - Modulation of the Host Innate Immune System. Front. Immunol. 2014, 5, 481, doi:10.3389/fimmu.2014.00481).
It is also possible to find that most authors published in journals of the American Society for Microbiology (www.asm.org), e.g., Infection and Immunity https://journals.asm.org/journal/iai, accept this role.
- L24: Bifidobacterium porcinum
Authors comment: Whole genus name is used in the text previously. It is permissible to use abbreviations further in the text.
- L25: Lactobacillus paracasei subsp. Tolerans
Authors comment: The bacterium name “L. paracasei subsp. tolerans“ used in the manuscript is correct and used in accordance with the principals of nomenclature.
- L55, 59: S. Typhimurium (italics)
Authors comment: The bacterium name „S. Typhimurium“ used in our manuscript is correct and used in accordance with the principles of nomenclature.
- L89: probiotic-defined pig
Authors comment: The used expression “probiotic defined pig microbiota” is correct. It is also used elsewhere in the text.
- L191: Table 2: full name of ‘’ S. warneri’’
Authors comment: corrected.
- L244: was verified on 1 day
Authors comment: corrected.
Reviewer 2 Report
Salmonella Typhimurium infection is a significant problem in both humans and animals.
In my opinion, the article presented to me for review is important from both a biological and practical and economic point of view. In the introduction, the authors clearly presented the issues related to the intestinal microbiota and infection with pathological bacteria ( Salmonella Typhimurium).
Remarkable are the modern research methods used in the work presented for review. The results of the study were presented in a clear and transparent form.
The authors used appropriate statistical methods.
In their work, the authors included 86 items of scientific bibliography.
The result of the work of the authors of the publication is the development of a mixture of 9 commensal bacteria with anti-Salmonella activity. As the authors noted in the conclusions, it will be necessary to conduct further studies under in vivo conditions to confirm the preliminary results of the study.
In my opinion, the work can be accepted for publication in its present form.
Author Response
Author comment: There are no comments which need to be addressed. Conclusions were improved.
Round 2
Reviewer 1 Report
No comments